# Raw Cow’s Milk Reduces Allergic Symptoms in a Murine Model for Food Allergy—A Potential Role for Epigenetic Modifications

**DOI:** 10.3390/nu11081721

**Published:** 2019-07-25

**Authors:** Suzanne Abbring, Johanna Wolf, Veronica Ayechu-Muruzabal, Mara A.P. Diks, Bilal Alashkar Alhamwe, Fahd Alhamdan, Hani Harb, Harald Renz, Holger Garn, Johan Garssen, Daniel P. Potaczek, Betty C.A.M. van Esch

**Affiliations:** 1Division of Pharmacology, Utrecht Institute for Pharmaceutical Sciences, Faculty of Science, Utrecht University, 3584 CG Utrecht, The Netherlands; 2Institute of Laboratory Medicine, Member of the German Center for Lung Research (DZL) and the Universities of Giessen and Marburg Lung Center (UGMLC), 35043 Marburg, Germany; 3College of Pharmacy, International University for Science and Technology (IUST), Daraa 15, Syria; 4Danone Nutricia Research, 3584 CT Utrecht, The Netherlands; 5John Paul II Hospital, 31-202 Krakow, Poland

**Keywords:** epigenetics, farming effect, food allergy, histone acetylation, milk processing, raw milk

## Abstract

Epidemiological studies identified raw cow’s milk consumption as an important environmental exposure that prevents allergic diseases. In the present study, we investigated whether raw cow’s milk has the capacity to induce tolerance to an unrelated, non-milk, food allergen. Histone acetylation of T cell genes was investigated to assess potential epigenetic regulation. Female C3H/HeOuJ mice were sensitized and challenged to ovalbumin. Prior to sensitization, the mice were treated with raw milk, processed milk, or phosphate-buffered saline for eight days. Allergic symptoms were assessed after challenge and histone modifications in T cell-related genes of splenocyte-derived CD4^+^ T cells and the mesenteric lymph nodes were analyzed after milk exposure and after challenge. Unlike processed milk, raw milk decreased allergic symptoms. After raw milk exposure, histone acetylation of Th1-, Th2-, and regulatory T cell-related genes of splenocyte-derived CD4^+^ T cells was higher than after processed milk exposure. After allergy induction, this general immune stimulation was resolved and histone acetylation of Th2 genes was lower when compared to processed milk. Raw milk reduces allergic symptoms to an unrelated, non-milk, food allergen in a murine model for food allergy. The activation of T cell-related genes could be responsible for the observed tolerance induction, which suggested that epigenetic modifications contribute to the allergy-protective effect of raw milk.

## 1. Introduction

Allergic diseases are a growing public health concern. In the previous decades, their prevalence has increased to such an extent that, nowadays, 20 to 30% of the world’s population is suffering from some form of allergic disease [1]. With a severe impact on quality of life and extensive healthcare costs, the vast prevalence of allergic diseases has major socio-economic consequences [2]. Unfortunately, to date, there is neither a cure nor an effective and safe treatment. Allergy management focuses on allergen avoidance and symptomatic treatment with the self-administration of epinephrine in the case of systemic anaphylaxis upon accidental exposure.

Even though there are no effective preventive approaches for allergic diseases, there seems to be a natural solution. Several epidemiological studies have shown that children growing up on a farm have a reduced risk of developing asthma and allergies compared to children living in the same rural area but not growing up on a farm [3,4,5,6,7]. This protective ‘farm effect’ was demonstrated in many populations and it persisted into adult life [8]. Farm exposures that were associated with this allergy-protective effect appeared to be contact with livestock and animal feed, exposure to stables and barns, and consumption of raw, unprocessed, cow’s milk [9,10,11]. Especially, the consumption of raw cow’s milk is of importance, since its protective effect was found to be independent of farm status, giving it the potential to confer protection for a general, non-farming, population [9,10,12,13]. Recently, these epidemiological findings were confirmed by showing a causal relationship between raw cow’s milk consumption and the prevention of allergic asthma in a murine model [14].

How raw cow’s milk can be allergy protective is currently still unclear. Neither the protective raw milk constituents nor the underlying mechanisms are known. Heat-sensitive milk components, like immunoglobulins, lactoferrin, alkaline phosphatase, TGF-β, microRNAs, etc., are likely candidates, since epidemiological as well as preclinical studies have shown that milk processing, and particularly heating, abolishes the allergy-protective effect of raw cow’s milk consumption [13,14,15,16]. However, the actual bioactive component(s) involved remain to be elucidated. Regarding the underlying mechanisms, several of the bioactive components that are present in raw milk are theoretically able to create a tolerogenic environment by, for example, promoting regulatory T cell development, enhancing epithelial barrier function and modulating the gut microbiome, however, none of these effects were actually investigated after drinking raw milk [17,18].

An emerging field is the contribution of epigenetic modifications in regulating the development of allergic diseases. Allergic diseases are the result of a complex interplay between the genes and environmental factors. These environmental factors can influence gene expression via epigenetic mechanisms, such as DNA methylation and histone modifications [19,20]. Epigenetic modifications are reversible and they affect the accessibility of the DNA to transcription enzymes, thereby regulating gene expression [19]. Environmental factors and components recently gaining interest in this regard are microbes, obesity, stress, and tobacco smoke, but it has also been suggested that nutrients might exert their effects through epigenetic mechanisms [19,21]. This indicates that epigenetic regulation might also be involved in the allergy-protective effect of raw cow’s milk consumption.

Before certified raw cow’s milk (raw cow’s milk obtained from a farm that is legally allowed to sell raw milk [22]) can become part of a preventive approach for allergic diseases, compelling evidence that thoroughly investigates components and mechanisms that are involved is needed. As a first step, the many epidemiological studies showing an allergy-protective effect of raw cow’s milk consumption need to be strengthened by causal evidence. In a previous study, we were able to show causality in a murine house dust mite-induced asthma model [14]. With the current research, we aimed to assess whether raw cow’s milk has the capacity to induce tolerance to an unrelated, non-milk, food allergen. Besides, we studied the contribution of epigenetic regulation by assessing histone acetylation of T cell-related genes, as a potential mechanism underlying the protective effects.

## 2. Materials and Methods

### 2.1. Animals

Specific pathogen-free, three- to five-week-old, female C3H/HeOuJ mice were purchased (Charles River Laboratories, Sulzfeld, Germany) and were randomly allocated to the control or experimental groups. The mice were housed in filter-topped makrolon cages (one cage/group, *n* = 6–8/cage) with standard chip bedding, Kleenex tissues, and a plastic shelter on a 12 h light/dark cycle with unlimited access to food (‘Rat and Mouse Breeder and Grower Expanded’; Special Diet Services, Witham, UK) and water at the animal facility of Utrecht University (Utrecht, The Netherlands). All animal procedures were approved by the Ethical Committee for Animal Research of the Utrecht University and were complied with the European Directive on the protection of animals used for scientific purposes (DEC 2014.II.12.107 & AVD108002015346).

### 2.2. Experimental Design—Tolerance Induction, Sensitization and Challenges

After an acclimatization period of one week, the mice were orally treated (i.e., intragastrically (i.g.) by using a blunt needle) with 0.5 mL certified raw, unprocessed, cow’s milk (Hof Dannwisch, Horst, Germany), processed shop milk (full fat milk, 3.5%; EDEKA, Germany), or phosphate-buffered saline (PBS; as a control) for eight consecutive days (days −9 to −2). Following this oral tolerance induction period, mice were sensitized i.g. once a week for five weeks to the hen’s egg protein ovalbumin (OVA; 20 mg/0.5 mL PBS; grade V; Sigma-Aldrich, Zwijndrecht, The Netherlands) while using 10 µg cholera toxin (CT; List Biological Laboratories, Campbell, CA, USA) as an adjuvant (days 0, 7, 14, 21, 28; *n* = 8/group). Sham-sensitized control mice (*n* = 6) received CT alone (10 µg/0.5 mL PBS). Five days after the last sensitization (day 33), all of the mice were intradermally (i.d.) challenged in both ear pinnae with 10 µg OVA in 20 µL PBS to determine acute allergic symptoms. Mice were subsequently i.g. challenged (7 h after the i.d. challenge) with 50 mg OVA in 0.5 mL PBS. Sixteen hours later (day 34), blood samples were taken via cheek puncture and mice were killed by cervical dislocation. The spleens were then collected for ex vivo analysis. Additional groups of mice (*n* = 6/group) were used in a follow-up experiment to assess the involvement of epigenetic regulation. These mice were killed by cervical dislocation either one day after the oral tolerance induction period (day −1) or one day after both challenges (day 34). Figure 1 shows a schematic representation of the experimental timeline.

### 2.3. Assessment of the Acute Allergic Response

The acute allergic skin response, anaphylactic shock symptoms, and body temperature were evaluated by a researcher blinded to treatment upon i.d. challenge with OVA (10 µg OVA/20 µL PBS) in the ear pinnae of both ears to determine the severity of the acute allergic symptoms. The acute allergic skin response was measured as Δ ear swelling (µm) by subtracting the mean ear thickness before i.d. challenge from the mean ear thickness 1 h after i.d. challenge. Ear thickness at both of the timepoints was measured in duplicate for each ear using a digital micrometer (Mitutoyo, Veenendaal, The Netherlands). The mice were anesthetized using inhalation of isoflurane to perform the i.d. challenge as well as the ear measurements (Abbott, Breda, The Netherlands). The severity of anaphylactic shock symptoms was determined 30 min after i.d. challenge by using a previously described, validated, scoring table [23]. Body temperature was also measured 30 min after i.d. challenge (using a rectal thermometer) to monitor the anaphylactic shock-induced drop in body temperature.

### 2.4. Detection of OVA-Specific IgE and mMCP-1 in Serum

Blood was collected via cheek puncture 16 h after i.g. challenge, centrifuged at 10,000 rpm for 10 min, and the serum was stored at −20 °C until the analysis of OVA-specific IgE and mouse mast cell protease-1 (mMCP-1) levels by means of ELISA. OVA-specific IgE titers were detected, as described previously [24]. Levels are expressed in arbitrary units (AU), which were calculated based on a titration curve of pooled sera serving as an internal standard. The concentrations of mMCP-1 were determined while using a mMCP-1 Ready-SET-Go!^®^ ELISA (eBioscience, Breda, The Netherlands), according to the manufacturer’s protocol.

### 2.5. Ex Vivo OVA-Specific Stimulation of Splenocytes for Cytokine Measurements

Spleens were collected and homogenized while using a syringe and a 70 µm nylon cell strainer. The obtained single cell splenocyte suspensions were incubated with lysis buffer (8.3 g NH_4_Cl, 1 g KHC_3_O and 37.2 mg EDTA dissolved in 1 L demi water, filter sterilized) to remove the red blood cells and then resuspended in RPMI 1640 medium (Lonza, Verviers, Belgium), supplemented with 10% heat-inactivated fetal bovine serum (FBS; Bodinco, Alkmaar, The Netherlands), penicillin (100 U/mL)/streptomycin (100 µg/mL; Sigma-Aldrich), and β-mercaptoethanol (20 µM; Thermo Fisher Scientific, Paisley, Scotland). The splenocytes (8 × 10^5^ cells/well) were cultured in U-bottom culture plates (Greiner, Frickenhausen, Germany), either with medium or with 50 µg/mL OVA for four days at 37 °C, 5% CO_2_. The supernatants were collected and stored at −20 °C until cytokine analysis. The concentrations of IL-5 and IL-13 were measured by means of ELISA, as described elsewhere [25]. The concentrations of IFNγ, IL-10, and IL-17 were measured using a Cytometric Bead Array (CBA) Mouse Th1/Th2/Th17 Cytokine Kit (BD Biosciences, Alphen aan de Rijn, The Netherlands), according to the manufacturer’s instructions. The results were obtained using FACS Canto II and analyzed with FCAP Array Software, version 3.0 (BD Biosciences, Alphen aan de Rijn, The Netherlands). Cytokine concentrations measured after medium stimulation were subtracted from cytokine concentrations measured after OVA stimulation to determine the OVA-specific cytokine response. A zero was entered when this resulted in a negative value.

### 2.6. Chromatin Immunoprecipitation to Determine Histone Acetylation Status in Splenocyte-Derived CD4^+^ T Cells and Mesenteric Lymph Nodes (MLN)

At day −1 (after the tolerance induction period) and at day 34 (after both challenges), CD4^+^ T cells were isolated from splenocytes of raw milk- and shop milk-treated mice using MACS, according to the manufacturer’s instructions (Miltenyi Biotec, Leiden, The Netherlands).

Isolated CD4^+^ T cells were frozen with 15% dimethyl sulphoxide (DMSO; Sigma-Aldrich) in heat-inactivated FBS (Bodinco) and then stored in liquid nitrogen until further analysis. For the MLN, the entire tissue, containing a full population of the MLN cells, was frozen in 15% DMSO-FBS and stored in liquid nitrogen until further analysis. Detailed methodology of chromatin immunoprecipitation, followed by real-time polymerase chain reaction (ChIP-qPCR), along with its thoughtful validations, were previously described in detail [26]. In brief, the MLN tissues were first smashed through a mesh, washed with 1 mL of PBS (Sigma-Aldrich), and centrifuged at 8000 rpm for 5 min. at 4 °C. The pellet was then resuspended in 1 mL of warm PBS. The cross-linking of the cells was performed by incubating the cells with paraformaldehyde (PFA; Carl Roth GmbH, Karlsruhe, Germany) to a final concentration of 1% for 8 min at room temperature. The reaction was quenched by adding glycine to a final concentration of 125 mM (Carl Roth GmbH). After centrifugation at 8000 rpm for 5 min at 4 °C and washing with cold PBS, the samples were subjected to 20 min of incubation with lysis buffer I (Appendix A) at 4 °C. Lysis buffer II (Appendix A) was added with 1% sodium dodecyl sulfate (SDS; Carl Roth GmbH) for 5 min at 4 °C. Shearing of the DNA-protein complexes with the Bioruptor (Diagenode, Liège, Belgium) was conducted afterwards while using 30 cycles (30 s on, 30 s off) for CD4^+^ T cells and 40 cycles (40 s on, 40 s off) for MLN cells. Finally, the interfering debris was removed by centrifugation at 15,000 rpm for 15 min at 4 °C. Sepharose beads (GE Healthcare Bio-Sciences, Uppsala, Sweden) were first washed with lysis buffer II with 0.1% SDS. Following centrifugation at 3000 rpm for 2 min at room temperature, the beads were blocked with 1 mg/mL bovine serum albumin (BSA; Sigma-Aldrich) and 40 µg/mL salmon sperm DNA (Sigma-Aldrich) overnight at 4 °C. After washing the prepared beads with lysis buffer II with 0.1% SDS and centrifugation at 3000 rpm for 5 min at 4 °C, 30 µL of beads slurry per immunoprecipitation (IP) per number of samples were stored at 4 °C for the next day. To perform chromatin preclearing, 20 µL of beads slurry per antibody were added to the previously cross-linked chromatin samples, incubated with rotation for 2 h at 4 °C, and then centrifuged at 8000 rpm for 5 min at 4 °C. To the rest of the beads, 500 µL of lysis buffer II with 0.1% SDS and 1 µg of unspecific IgG (Abcam, Cambridge, UK) per sample were added and then incubated with rotation for 1 h at 4 °C. After washing three times with lysis buffer II with 0.1% SDS, 20 µL of the IgG-coupled beads were added to the precleared chromatin, incubated with rotation for 2 h at 4 °C, and then centrifuged at 8000 rpm for 5 min at 4 °C. Ten percent of the resulting supernatant containing chromatin were stored as the input control. The rest was divided into equal parts, to which 4 µg of either H3 or H4 (Millipore, Darmstadt, Germany) or 0.5 µg of IgG (Abcam) were added. The samples were then incubated at 4 °C overnight. Thirty microliter of the blocked beads slurry kept aside before, were added to each IP, and incubated for 2 h at 4 °C. After centrifugation at 8000 rpm for 5 min at 4 °C, the beads were washed twice with wash buffer I, twice with lysis buffer II, three times with wash buffer III (Appendix A), and then twice with TE buffer with pH 8.0 (Appendix A). The elution of the chromatin was performed by adding 500 µL of elution buffer (Appendix A) to the sepharose beads, vortexing and incubating with rotation for 30 min After centrifugation at 8000 rpm for 2 min at 4 °C, the supernatants containing each IP, as well as the input controls, were mixed with 20 µL of 5 M NaCl, 10 µL of 0.5 M EDTA (Sigma-Aldrich), 20 µL of 1 M Tris-HCl (pH 7.2), 1 µL of Protease K (20 mg/mL; Sigma-Aldrich), and 1 µL of RNAse A (10 mg/mL; Sigma-Aldrich) per sample. All of the samples were incubated at 55 °C for 3 h and then at 65 °C overnight. Afterwards, DNA was purified while using the QIAquick PCR purification kit (Qiagen, Hilden, Germany). The purified DNA was subjected to qPCR that was performed with specific mouse gene promoter primers (Appendix A) and Rotor-Gene SYBR Green PCR Kit (Qiagen), performed on Rotor-Gene Q (Qiagen). We were unfortunately unable to successfully amplify RORγ from H4-immunoprecipitated MLN DNA despite of two rounds of repetition, most probably due to the presence of a specific inhibition of this PCR in this batch of the samples. Percent enrichment to the input was calculated using the following formula: ${\%\;\mathrm{enrichment}\;=\;100\;\times \;2}^{\left[\left(\mathrm{CT}\;\mathrm{input}-3.3\right)-\mathrm{CT}\;\mathrm{sample}\right]}$. Subsequently, the % enrichment of the isotype (IgG) control was subtracted from % enrichments that were obtained for specific antibodies. For final normalization, to further eliminate the variation caused by sample handling, such value obtained for each specific gene was divided by that of the positive control gene ribosomal protein L32 (RPL32) [26,27].

### 2.7. Statistical Analysis

Experimental results are expressed as mean ± standard error of the mean or as individual data points or box-and-whisker Tukey plots when the data were not normally distributed and analyzed using GraphPad Prism software (version 7.03, GraphPad Software, San Diego, CA, USA). Differences between pre-selected groups were statistically determined using one-way ANOVA, followed by Bonferroni’s multiple comparisons test. Square root transformation was applied to mMCP-1 concentrations prior to ANOVA analysis. Anaphylactic shock scores and OVA-specific IgE levels were analyzed using the Kruskal–Wallis test for non-parametric data, followed by Dunn’s multiple comparisons test for pre-selected groups. For histone acetylation and cytokine concentrations, differences between groups were statistically determined with an unpaired two-tailed Student’s t-test. Welch’s correction was used when the group variances were not equal. When data did not obtain normality, a Mann–Whitney test was performed. The results were considered to be statistically significant when *p* < 0.05.

## 3. Results

### 3.1. Raw Milk Reduces OVA-Induced Allergic Symptoms

Mice were orally treated for eight consecutive days with raw, unprocessed, cow’s milk before being sensitized and challenged with OVA to assess whether raw cow’s milk has the capacity to induce tolerance to an unrelated, non-milk, food allergen. Upon i.d. challenge with OVA, acute allergic symptoms were, as expected, increased in OVA-sensitized allergic mice when compared to PBS-sensitized control mice. An increased acute allergic skin response, increased anaphylactic shock symptoms, and an anaphylactic shock-induced drop in body temperature illustrated this (Figure 2A–C). Treating mice with raw milk prior to OVA-sensitization reduced acute allergic symptoms when compared to PBS-treated allergic mice. The allergic skin response and anaphylactic shock symptoms were decreased and the body temperature of these mice remained high (Figure 2A–C). Mice were also treated with a processed, shop, milk to determine whether this allergy-suppressive effect is abolished upon milk processing. Treatment with this shop milk did not confer protection against allergic symptoms (Figure 2A–C).

### 3.2. OVA-Specific IgE Levels and Mucosal Mast Cell Degranulation Are Not Affected by Raw Milk Exposure

The effect of raw and shop milk on serum OVA-IgE levels was investigated since food allergens mainly induce type I hypersensitivity reactions, which are characterized by the production of allergen-specific IgE antibodies. Serum OVA-IgE levels were elevated in OVA-sensitized mice when compared to PBS-sensitized mice (Figure 3A). Even though OVA-IgE levels were not significantly affected by exposure to both milk types, they did follow a similar pattern as the acute allergic symptoms, with low OVA-IgE levels in the raw milk group and higher levels in the shop milk group (Figure 3A). In addition, serum mMCP-1 concentration, as a marker for mucosal mast cell degranulation, was measured. mMCP-1 concentrations were increased in the OVA-sensitized mice when compared to PBS-sensitized mice, but were unaffected by treatment with raw or shop milk (Figure 3B).

### 3.3. Raw Milk Treatment Initially Increases Histone Acetylation of Several T Cell Subset Genes, While after Both Challenges It Specifically Reduces Th2-Related Gene Acetylation

Environmental factors might interact with genes that are involved in allergy development via epigenetic regulation. Histone acetylation (associated with higher gene expression) at selected Th1-, Th2-, Th17-, and regulatory T cell (Treg)-specific genes of splenocyte-derived CD4^+^ T cells was assessed to determine whether epigenetic modifications contribute to the allergy-protective effect of raw cow’s milk consumption. Surprisingly, histone H4 acetylation of Th2-related genes (GATA3, IL-4, IL-5, and IL-13) was higher after eight days of raw milk exposure when compared to shop milk exposure (day −1; Figure 4A). Raw milk exposure also increased the histone acetylation of T-bet and tended to increase the histone acetylation of FoxP3 (day −1), which indicated a type of general immune stimulation (Figure 4D). After both challenges (day 34), this general immune stimulation that was induced by raw milk was resolved and the histone acetylation of Th2 genes was lower as compared to shop milk (Figure 4B,E). Furthermore, the histone acetylation pattern of Th2-related genes is visualized by the raw milk/shop milk ratio, which shifted from in favor of raw milk after tolerance to in favor of shop milk after challenge (Figure 4C). A similar pattern was observed for IL-17, whereas the raw milk/shop milk ratio for Th1- and Treg-specific genes remained in favor of raw milk throughout the experiment (Figure 4F). For histone H3, the acetylation patterns were comparable (Appendix A).

### 3.4. Systemically Observed Acetylation Profile of Th2-Related Genes Induced by Raw Milk also Visible Locally

MLN were analyzed to determine whether the systemically observed alterations in histone H4 acetylation of T cell genes induced by raw milk are also visible locally. Despite being less strong, the shift in acetylation of Th2-related genes was also evident in the MLN (Figure 5A–C). Raw milk exposure for eight days led to higher acetylation of Th2-related cytokine genes (IL-4, IL-5, and IL-13) when compared to shop milk (day −1), while a lower acetylation of these genes was observed after both challenges (day 34; Figure 5A,B). For GATA3, histone acetylation was lower in the raw milk group after tolerance, as well as after the challenges (Figure 5A,B). The general immune stimulation, as observed after tolerance in CD4^+^ T cells derived from the spleen of raw milk-treated mice, was not observed in the MLN. No significant differences were found between raw milk and shop milk in histone acetylation levels at Th1, Th17, and Treg loci (Figure 5D). After the challenges, histone acetylation of T-bet was increased in shop milk-treated mice when compared to raw milk-treated mice (Figure 5E), which resulted in a shift in the raw milk/shop milk ratio towards more favorable in shop milk after challenge (Figure 5F). A similar shift was observed for IL-10 (Figure 5F). Histone H3 acetylation was also assessed for MLN, but no significant differences between the groups were observed (Appendix A).

### 3.5. Cytokine Production by OVA-Stimulated Splenocytes Corresponds to Histone Acetylation

Cytokine production upon ex vivo stimulation of splenocytes with OVA was measured since differences in histone acetylation levels of cytokine genes do not necessarily result in differences in actual cytokine production. To be able to look at the OVA-specific cytokine response, the concentrations were only measured after both challenges (day 34). Concentrations were low for the Th2-related cytokines IL-5 and IL-13 (Figure 6A,B). However, the tendency towards a reduced IL-5 production in raw milk-treated mice is of interest when compared to shop milk-treated mice (Figure 6A), which corresponds to the lower IL-5 acetylation in splenocyte-derived CD4^+^ T cells that were observed in histones H4 and H3 (Figure 4B and Appendix A). IFNγ and IL-17 concentrations also correspond with the observed acetylation patterns, although no significant difference between the milk groups was observed (Figure 4E and Figure 6C,E). In the case of IL-10, the cytokine concentration did not resemble gene acetylation, since the reduced IL-10 production in raw milk-treated mice was not observed in IL-10 gene acetylation (Figure 4E and Figure 6D). Ex vivo stimulation of MLN with OVA did not result in measurable cytokine production.

## 4. Discussion

After showing causality in a murine house dust mite-induced asthma model [14], the present study demonstrates that raw, unprocessed, cow’s milk is also protective in a murine model for food allergy. Raw milk induced oral tolerance to a non-milk, food allergen, by reducing acute allergic symptoms after intradermal challenge with OVA. This protective effect was not observed when a processed, shop milk was used to treat the mice. Looking at epigenetic modifications, raw milk exposure for eight days prior to sensitization led to higher histone acetylation of Th1-, Th2-, and Treg-related genes of splenocyte-derived CD4^+^ T cells when compared to shop milk exposure. At the end of the study, after the induction of allergic symptoms, this general immune stimulation was resolved and histone acetylation of Th2-related genes was lower when compared to shop milk. A similar, but less strong, pattern was locally visible, in the MLN. These results suggest that epigenetic regulation plays a role in the allergy-protective effect of raw milk.

Food allergies are thought to occur due to the failure to develop or the loss of oral tolerance [28]. Oral tolerance is the phenomenon of local and systemic immune hyporesponsiveness to ingested food proteins [29]. Actively inducing or restoring oral tolerance is an interesting approach for preventing or treating food allergies. For this, research has mainly focused on specific immunomodulation while using the allergen. Both inducing oral tolerance by allergen exposure in early life and restoring oral tolerance via various types of allergen-specific immunotherapy are frequent topics of immunological research [30,31]. However, using the intact allergen for oral tolerance induction might also trigger sensitization or allergic symptoms in high-risk patients [32,33].

Instead of specific immunomodulation, generic immunomodulation does not use the allergen to induce oral tolerance, preventing the risk of severe side effects. Generic immunomodulation is based on using beneficial immunomodulatory components that can create an environment that favors oral tolerance induction [34]. Mainly dietary components, such as, probiotics, prebiotics, synbiotics, and n-3 polyunsaturated fatty acids (PUFAs) have proven to be beneficial in this respect [35].

Several epidemiological studies already suggested that raw, unprocessed, cow’s milk may have the capacity to prevent allergic diseases by inducing tolerance via generic immunomodulation. Raw cow’s milk consumption was, for example, shown to be inversely associated with asthma, which indicated protection in the absence of the allergen [13]. In a murine house dust mite-induced asthma model, we confirmed these findings by showing a causal relationship between raw cow’s milk consumption and the prevention of allergic asthma [14]. In the current study, raw cow’s milk induced tolerance to OVA, an unrelated, non-milk, food allergen, which further substantiates this hypothesis.

Strikingly, processed, shop milk was not able to induce tolerance to OVA. This confirms earlier findings, which showed that milk processing abolishes the allergy-protective effect of raw milk [13,14,15,16]. The milk processing chain consists of various steps to preserve milk along the supply chain. Each of these steps (e.g., machine milking, skimming, homogenization, heat treatment, storage, and packaging) induces changes in the composition of the milk, which makes it hard to pinpoint one particular raw milk constituent that is responsible for the protective effects [36]. Even though comparing a raw milk with a shop milk (consumed by most people) was a logical first step in our opinion, future research should focus on testing milk from the same milk source that only differs in one processing step (skimmed milk, pasteurized milk, ultra-high temperature processing milk, etc.). Besides elucidating the raw milk component(s) involved, this will give the opportunity to look into the cellular mechanisms inducing tolerance in more depth.

Epigenetic regulation might be one of the mechanisms by which raw cow’s milk exerts its allergy-protective effect. Since environmental factors are known to be able to modulate gene expression through epigenetic mechanisms, we wondered whether this also applied to raw milk. Epigenetic mechanisms can modify the accessibility of genes for transcription without altering the DNA nucleotide sequence which means that they can modulate the phenotype without affecting the genotype [19]. In this way, epigenetic mechanisms are key in the plasticity of gene expression. They are essential for developmental processes, like cellular differentiation, contributing, for example, to the flexibility among CD4^+^ T cell subsets [37]. The classical epigenetic mechanisms comprise DNA methylation and histone modifications, including histone acetylation, methylation, phosphorylation, and ubiquitination [20].

We assessed histone acetylation at the promoter regions of Th1-, Th2-, Th17-, and Treg-related genes of splenocyte-derived CD4^+^ T cells and MLN to determine the role of epigenetic mechanisms in the allergy-protective effect of raw milk. During histone acetylation, an acetyl group is added to a lysine residue at the N-terminal tail of a histone (mainly histones H3 and H4). This removes the positive charge on the histones that are involved, resulting in a decreased interaction with the negatively charged DNA. Consequently, the DNA is less tightly wrapped around the histones, which makes it more accessible to the transcriptional machinery. Therefore, higher histone acetylation usually results in higher gene transcription, while the opposite is true for reduced histone acetylation [19].

In line with the protective effects that were observed on acute allergic symptoms and IgE, histone acetylation of Th2-related genes (GATA3, IL-4, IL-5, and IL-13) of splenocyte-derived CD4^+^ T cells after allergy induction was lower in raw milk-treated mice than in shop milk-treated mice. The strongest effects were observed on histone H4 acetylation at Th2 cytokine genes. Since histone acetylation substantially contributes to and is an important marker for an open chromatin structure [19,20], we assessed whether the acetylation levels positively correlated with cytokine production. Unfortunately, Th2 cytokine concentrations were low, but the tendency towards a reduced IL-5 production in raw milk-treated mice as compared to shop milk-treated mice suggests that there is indeed a positive correlation. Several other studies already confirmed that differences in H4 acetylation levels at Th2 cytokine genes indeed correlate with cytokine production [26,38]. Affecting epigenetic marks on Th2 cytokine genes might be an interesting preventive approach since type 2 cytokines play a predominant role in allergic diseases by directing the effector phase of an allergic response [39].

After allergy induction, the histone acetylation of Th1-, Th17-, and Treg-related genes did not differ between raw milk- and shop milk-treated mice. Although, here, histone acetylation patterns were reflected in cytokine production. The only cytokine for which the production did not correspond to gene acetylation was IL-10, which suggested that histone H3/H4 acetylation is not a main driver of IL-10 synthesis. Furthermore, we observed that IL-10 production was reduced in raw milk-treated mice as compared to shop milk-treated mice. This seems to be in contrast with the observed allergy protection, since IL-10 is known as a regulatory cytokine. However, in a murine model for OVA-induced food allergy, it was shown that IL-10 could also have proinflammatory effects. IL-10 was demonstrated to be essential for the development of food allergy by inducing mucosal mast cell expansion and activation [40]. This indicates that lowering IL-10 concentrations in a murine OVA-induced food allergy model might be beneficial. Besides systemically looking at splenocyte-derived CD4^+^ T cells, we also locally assessed histone acetylation in the MLN. Here, similar effects were observed, although less strong. This might have to do with the fact that the whole tissue was used for ChIP analysis, rather than the isolated T cells. This may have resulted in weaker effects, as other cell types might also express the genes measured.

In addition to looking at histone acetylation patterns at the end of the study (after allergy induction), we also directly assessed histone acetylation after the eight days of milk exposure. Surprisingly, histone acetylation of the Th2-related genes of splenocyte-derived CD4^+^ T cells was higher in the raw milk group as compared to the shop milk group. However, histone acetylation of T-bet and FoxP3 was also increased, which suggested a kind of general immune stimulation. Whether this general immune stimulation induced by raw milk is responsible for the observed allergy protection at the end of the study we do not know yet. Previously, however it has been demonstrated that acquiring tolerance in food allergic children involves epigenetic regulation of the FoxP3 gene [41]. Furthermore, epidemiological studies have shown that raw cow’s milk consumption was associated with increased DNA demethylation of FoxP3 and increased numbers of Tregs [42]. Unfortunately, we did not look at Treg numbers in our study, but since active suppression by Tregs is considered to be one of the main effector mechanisms for oral tolerance [43], the observed increase in histone acetylation of the FoxP3 gene might contribute to the allergy-protective effect. Inhibiting de novo histone acetylation with histone acetyltransferase inhibitors might be an interesting approach for further investigating the role of histone acetylation in the allergy-protective effect of raw milk.

How raw milk affects epigenetic marks on T cell-related genes is currently unclear, but there are some indications. Microbes that were derived from farm dust, known to prevent allergic asthma, were, for example, shown to operate via epigenetic mechanisms [44], which suggested that microbes that are present in raw milk might have similar effects. Furthermore, raw milk contains higher levels of n-3 PUFAs than industrially processed milk [15]. These n-3 PUFAs reduce the risk of developing allergic diseases and they have been shown to lower the acetylation of IL-13 genes [45,46]. In addition, raw milk contains components, like lactoferrin, which can promote the growth of *Bifidobacteria* and *Lactobacilli* in the gut [17,18]. These bacteria are potent producers of short-chain fatty acids and these short-chain fatty acids are known for their capacity to inhibit histone deacetylases, thereby increasing gene transcription. Whether the above-mentioned components in the concentrations present in raw milk can influence epigenetic mechanisms and subsequently contribute to the allergy-protective effect of raw milk should be clarified in future studies. The possible involvement of the epigenetic mechanisms should also be investigated in the case of the anti-allergic effects of human breast milk consumption [47].

## 5. Conclusions

In conclusion, we show the potency of raw cow’s milk to induce tolerance to a non-milk, food allergen. This allergy-protective effect was abolished by industrial milk processing, emphasizing the importance of minimally processed milk. The allergy-protective constituents of raw milk remain elusive and it should be investigated in follow-up studies. In addition, we showed that raw milk is able to modulate gene expression through epigenetic mechanisms. Raw milk might have induced oral tolerance by targeting histone marks on T cell-related genes. Whether this is a cause–effect relationship and whether effects are more pronounced with longer raw milk exposure should be assessed in future research. Nevertheless, our data suggest that the consumption of certified raw cow’s milk can contribute to allergy prevention and epigenetic regulations, especially histone modifications, might be one of the underlying mechanisms.

## Figures and Tables

**Figure 1 nutrients-11-01721-f001:**
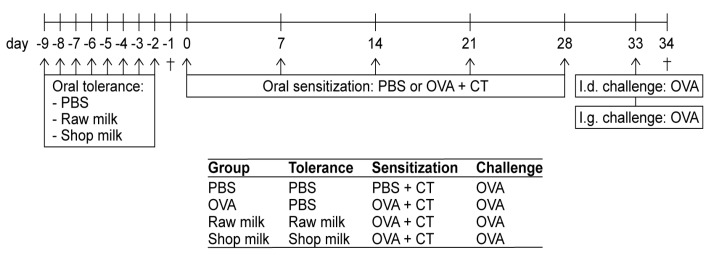
Schematic representation of the experimental timeline. For epigenetic measurements, additional groups of mice were killed after the tolerance induction period (day −1) and after both challenges (day 34; as indicated by †). PBS, phosphate-buffered saline; OVA, ovalbumin; CT, cholera toxin; i.d., intradermal; i.g., intragastric.

**Figure 2 nutrients-11-01721-f002:**
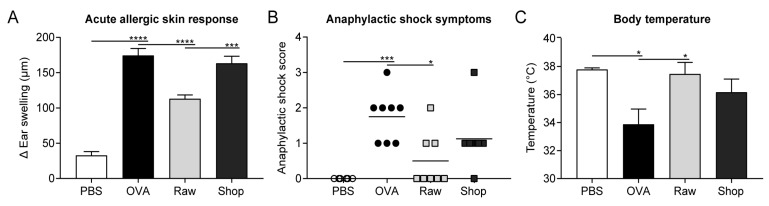
Reduced acute allergic symptoms upon ovalbumin (OVA) challenge in mice treated with raw milk. (**A**) The acute allergic skin response measured as Δ ear swelling 1 h after intradermal (i.d.) challenge. (**B**) Anaphylactic shock scores and (**C**) body temperature determined 30 min after i.d. challenge. Data are presented as mean ± standard error of the mean for the acute allergic skin response and body temperature and as individual data points for anaphylactic shock scores, *n* = 6 in PBS group and *n* = 8 in all other groups. * *p* < 0.05, *** *p* < 0.001, **** *p* < 0.0001, as analyzed with one-way ANOVA followed by Bonferroni’s multiple comparisons test for pre-selected groups (**A**,**C**) or Kruskal–Wallis test for non-parametric data followed by Dunn’s multiple comparisons test for pre-selected groups (**B**). PBS, phosphate-buffered saline; OVA, ovalbumin; raw, raw cow’s milk; shop, shop milk.

**Figure 3 nutrients-11-01721-f003:**
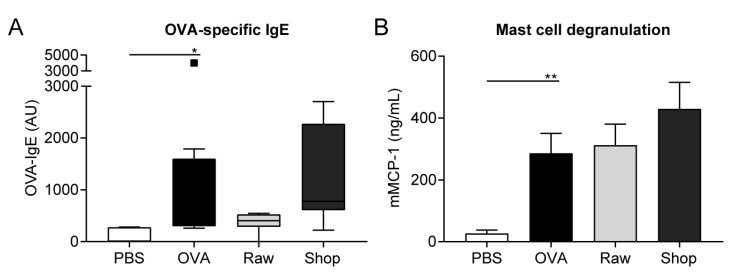
Raw milk treatment did not affect ovalbumin (OVA)-specific IgE levels and mouse mast cell protease-1 (mMCP-1) concentrations. (**A**) OVA-specific IgE levels and (**B**) mMCP-1 concentrations measured in serum 16 h after intragastric challenge. Data are expressed as box-and-whisker Tukey plot (in which outliers are shown as separately plotted points) for OVA-specific IgE levels and as mean ± standard error of the mean for mMCP-1 concentrations, *n* = 6 in PBS group and *n* = 8 in all other groups. * *p* < 0.05, ** *p* < 0.01 as analyzed with Kruskal–Wallis test for non-parametric data followed by Dunn’s multiple comparisons test for pre-selected groups (**A**) or one-way ANOVA followed by Bonferroni’s multiple comparisons test for pre-selected groups (**B**). PBS, phosphate-buffered saline; OVA, ovalbumin; AU, arbitrary units; raw, raw cow’s milk; shop, shop milk; mMCP-1; mucosal mast cell protease-1.

**Figure 4 nutrients-11-01721-f004:**
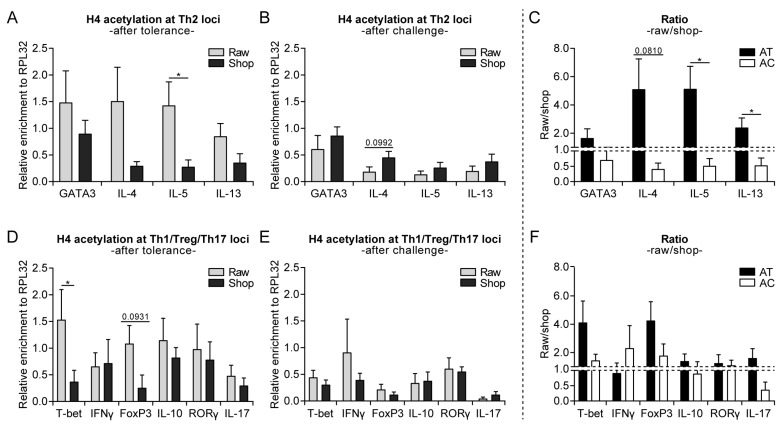
Increased histone acetylation of several T cell subset genes directly after raw milk exposure, while only Th2-related gene acetylation was reduced in raw milk-treated mice after both challenges. (**A**) Histone H4 acetylation at Th2 loci after the tolerance induction period (day −1), (**B**) after both challenges (day 34) and (**C**) the raw milk/shop milk ratio. (**D**) Histone H4 acetylation at Th1/Treg/Th17 loci after the tolerance induction period (day −1), (**E**) after both challenges (day 34) and (**F**) the raw milk/shop milk ratio. Histone H4 acetylation status was determined by means of chromatin immunoprecipitation in CD4^+^ T cells derived from splenocytes of raw milk and shop milk-treated mice. Results are expressed as relative enrichment after normalization to ribosomal protein L32 (RPL32) as mean ± standard error of the mean, *n* = 6/group. * *p* < 0.05 as analyzed with an unpaired two-tailed Student’s *t*-test. A Mann–Whitney test was used for T-bet, IFNγ, FoxP3, RORγ (after tolerance), T-bet, IL-17 (after model), and T-bet, IFNγ, RORγ (ratio raw/shop) since data did not obtain normality. Raw, raw cow’s milk; shop, shop milk; AT, after tolerance; AC, after challenge.

**Figure 5 nutrients-11-01721-f005:**
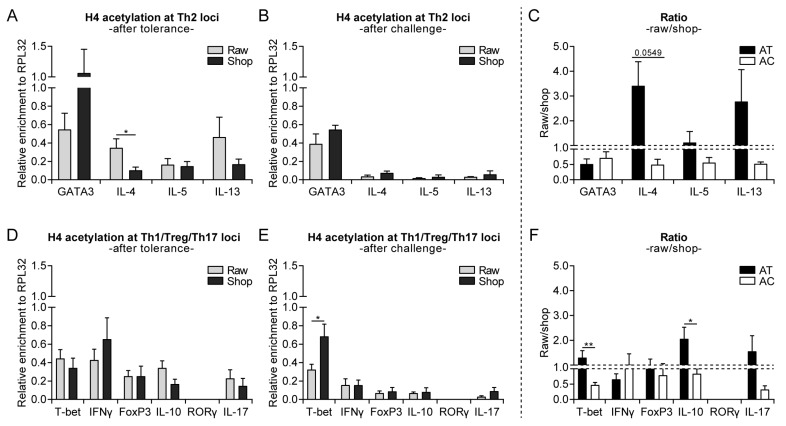
Raw milk-induced acetylation pattern of Th2-related genes observed in splenocyte-derived CD4^+^ T cells also visible locally in mesenteric lymph nodes (MLN). (**A**) Histone H4 acetylation at Th2 loci after the tolerance induction period (day −1), (**B**) after both challenges (day 34) and (**C**) the raw milk/shop milk ratio. (**D**) Histone H4 acetylation at Th1/Treg/Th17 loci after the tolerance induction period (day −1), (**E**) after both challenges (day 34) and (**F**) the raw milk/shop milk ratio. Histone H4 acetylation status was determined by means of chromatin immunoprecipitation in MLN of raw milk- and shop milk-treated mice. The results are expressed as relative enrichment after normalization to ribosomal protein L32 (RPL32) as mean ± standard error of the mean, *n* = 4–6/group. * *p* < 0.05, ** *p* < 0.01 as analyzed with an unpaired two-tailed Student’s *t*-test. A Mann–Whitney test was used for GATA3, IL-10 (after tolerance), IL-10 (after model) and GATA3 (ratio raw/shop) since data did not obtain normality. Raw, raw cow’s milk; shop, shop milk; AT, after tolerance; AC, after challenge; MLN; mesenteric lymph nodes.

**Figure 6 nutrients-11-01721-f006:**
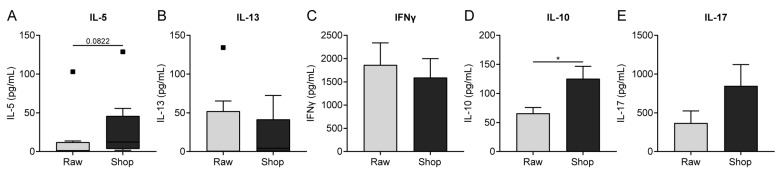
Cytokine concentrations produced by ovalbumin (OVA)-stimulated splenocytes corresponded with observed histone acetylation. (**A**) IL-5, (**B**) IL-13, (**C**), IFNγ, (**D**) IL-10 and (**E**) IL-17 concentrations measured in supernatant after ex vivo stimulation of splenocytes with OVA for four days (37 °C, 5% CO_2_). Data are presented as box-and-whisker Tukey plot (in which outliers are shown as separately plotted points) for IL-5 and IL-13 concentrations and as mean ± standard error of the mean for IFNγ, IL-10 and IL-17 concentrations after subtracting baseline cytokine levels, *n* = 8/group. * *p* < 0.05 as analyzed with a Mann-Whitney test (**A**,**B**) or an unpaired two-tailed Student’s *t*-test (**C**–**E**). OVA, ovalbumin; raw, raw cow’s milk; shop, shop milk.

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
