# Peer review of "Raw Cow’s Milk Reduces Allergic Symptoms in a Murine Model for Food Allergy—A Potential Role for Epigenetic Modifications"

_nutrients, 2019, doi:10.3390/nu11081721_

Reviewer 1 Report

This is a great manuscript that further dives into the mechanisms behind the relationship between consumption of raw milk and decreased incidence of allergic disorders

As epigenetic modifications in regulating the development of allergic diseases is being explored, this paper studies the contribution of epigenetic regulation by assessing histone acetylation of T cell-related genes that may play a role in the protective effects in mouse models.

It was a well written manuscript and I do not have any additional suggestions for the authors.

Author Response

The authors would like to thank reviewer 1 for his/her time and effort to carefully read the manuscript. We also would like to thank reviewer 1 for the kind words about our manuscript.

Reviewer 2 Report

The manuscript “Raw cow’s milk suppresses allergic symptoms in a murine model for food allergy - a potential role for epigenetic modifications” by Suzanne Abbring and collaborators describes a study aimed at assessing the protective effects of raw cow’s milk on allergic disease. The role of epigenetic modifications, especially histone acetylation, in T cell-related genes of splenocyte derived CD4+ T cells and mesenteric lymphonodes in a murine model for food allergy was also investigated.

General comments

Although we can appreciate the Authors effort in performing so many experiments, the results here reported lack of originality. In fact, many published research papers already describe the allergy-protective effect of raw cow’s milk. Therefore, the real goal would be the identification of the bioactive components responsible for the protective effects. In addition, the study is not well designed and the results do not support the conclusions.

Some specific comments

-Title. The title of this paper states that “Raw cow’s milk suppresses allergic symptoms in …” Indeed, the Results show a reduction of symptoms and not a suppression. Therefore, the title is not correct.

-The authors compare the results obtained with raw cow’s milk with those obtained with a different milk (commercial). Then they state that the raw milk has protective effects. They should compare the results obtained with the raw milk with those obtained with the same milk after processing.

Author Response

Reviewer 2:

The authors would like to thank reviewer 2 for the comments on our manuscript. We address the questions raised by reviewer 2 by answering the specific comments point-by-point.

General comments

Although we can appreciate the Authors effort in performing so many experiments, the results here reported lack of originality. In fact, many published research papers already describe the allergy-protective effect of raw cow’s milk. Therefore, the real goal would be the identification of the bioactive components responsible for the protective effects. In addition, the study is not well designed and the results do not support the conclusions.

We hope reviewer 2 appreciates the fact that the many published papers describing an allergy-protective effect of raw cow’s milk are all epidemiological studies. The observed associations in these studies do not confirm a causal relationship. Without proof of causality, the potential allergy-protective effects of raw cow’s milk will always be questioned. Controlled intervention studies to show causality have not been conducted yet and preclinical models offer a solution.

We previously strengthened the epidemiological evidence by demonstrating a causal relationship between raw cow’s milk consumption and the prevention of house dust mite-induced allergic asthma in a murine animal model*. In the current study we showed that raw cow’s milk is also able to reduce allergic symptoms in a murine model for food allergy. Besides, we tried to get more insight in the potential mechanism underlying the protective effects of raw cow’s milk by studying the contribution of epigenetic regulation. We therefore believe that the present study provides originality.

We do agree with reviewer 2 that elucidating the bioactive raw milk component(s) responsible for the allergy-protective effects is the next step to be taken. This was therefore addressed by a follow-up study, which is recently published, in which we demonstrated that not the fat content, but the heat-sensitive raw milk components are responsible for the observed protection against food allergic symptoms**.

* Abbring, Suzanne, et al. "Raw cow’s milk prevents the development of airway inflammation in a murine house dust mite-induced asthma model." Frontiers in immunology 8 (2017): 1045.
**
Abbring, Suzanne, et al. "Suppression of Food Allergic Symptoms by Raw Cow’s Milk in Mice is Retained after Skimming but Abolished after Heating the Milk—A Promising Contribution of Alkaline Phosphatase." Nutrients 11.7 (2019): 1499.

Some specific comments

-Title. The title of this paper states that “Raw cow’s milk suppresses allergic symptoms in …” Indeed, the Results show a reduction of symptoms and not a suppression. Therefore, the title is not correct.

We agree with reviewer 2 that ‘reduces’ might better cover the content of the manuscript.
We therefore changed the title of the manuscript to ‘Raw cow’s milk reduces allergic symptoms in a murine model for food allergy – A potential role for epigenetic modifications’ (page 1, line 2).

-The authors compare the results obtained with raw cow’s milk with those obtained with a different milk (commercial). Then they state that the raw milk has protective effects. They should compare the results obtained with the raw milk with those obtained with the same milk after processing.

In this first study on the capacity of raw cow’s milk to protect against OVA-induced allergic symptoms in a murine model, we decided to compare raw cow’s milk with a commercially available shop milk which is consumed by most of the people. In our opinion it was a logical step to compare two extremes in this first preclinical study on food allergy; a raw milk which did not receive any treatment and a store-bought milk which is extensively processed. Although we agree that these milk types differ in many respects, the results do show that only raw milk is able to reduce allergic symptoms. The milk that is currently consumed by many people did not show this protective capacity.  
We agree with reviewer 2 that the ultimate control would be a milk from the same milk source differing in only one processing step from the raw milk used. This was therefore already stated in the discussion section as a limitation of the current study (page 10, line 383-386).
In a follow-up study*, which was recently published, we compared raw milk with a skimmed milk and a pasteurized milk obtained from the same milk source to get more insight in the raw milk components involved in the allergy protective effect. Here, we demonstrated that the reduction of food allergic symptoms by raw cow’s milk was retained after skimming but abolished after pasteurization of the milk, indicating a promising role for heat-sensitive raw milk components.

* Abbring, Suzanne, et al. "Suppression of Food Allergic Symptoms by Raw Cow’s Milk in Mice is Retained after Skimming but Abolished after Heating the Milk—A Promising Contribution of Alkaline Phosphatase." Nutrients 11.7 (2019): 1499.

Reviewer 3 Report

This manuscript describes a detailed investigation into the differential effects of raw milk versus shop (processed) milk on the induction and manifestation of ovalbumin (food) allergy. Using a murine model for food allergy (ovalbumin), the authors have convincingly demonstrated the effects of raw or processed milk on the allergic symptoms upon challenge, humoral response, cytokine response and most interestingly, the potential underlying epigenetic changes associated with the differences in the allergic response. This is a well-designed study, demonstrating the beneficial effects of raw milk consumption, in its protective effects against food allergy. However, the precise component of raw milk (that is not present in processed shop milk) has yet to be elucidated.

The manuscript is written in good detail, and I have minor suggestions and comments,

1)     The methods section 2.6 to determine histone acetylation status is quite detailed. However, as a non-specialist in epigenetic analysis, it was not clear what the difference in analysis were for CD4+ T cells and the MLN tissue cells. A bit of clarification and explanation on why do you look at relative expression to RPL32 gene would be useful for the readers.

2)     Is there any explanation for the unchanged levels of Ova sIgE and mast cell degranulation? Has this phenomena been observed before in similar studies?

3)     Out of curiosity, why was only acetylation of histones analysed as a marker of epigenetic changes and not the other modifications?

4)     From the outcomes of this study, especially with the epigenetic changes before and after challenge, would you be able to comment how this can translate into human milk consumption that may have protective effects against food allergy?

Author Response

Reviewer 3:

The manuscript is written in good detail, and I have minor suggestions and comments,

1)     The methods section 2.6 to determine histone acetylation status is quite detailed. However, as a non-specialist in epigenetic analysis, it was not clear what the difference in analysis were for CD4+ T cells and the MLN tissue cells. A bit of clarification and explanation on why do you look at relative expression to RPL32 gene would be useful for the readers.

Thank you very much for these comments. We apologize for this lack of clarity. While in case of spleen, pure CD4+ T cells were isolated, the MLN tissue was analyzed as a whole, with all cells included/no cell subpopulation isolated.

We have expanded the respective sentence accordingly as follows: “For the MLN, the entire tissue, containing a full population of the MLN cells, was frozen in 15% DMSO-FBS and stored in liquid nitrogen until further analysis.” (page 4, line 161-162).

Regarding the normalization to RPL32 gene, this additional measure of normalization was used in this study to further eliminate variation caused by sample handling*.

We have added this piece of information to the respective sentence, as follows: “For final normalization, to further eliminate variation caused by sample handling, such value obtained for each specific gene was divided by that of the positive control gene RPL32, [Haring et al., 2007; Harb et al., 2015.” (page 5, line 208-210).

*Haring, Max, et al. "Chromatin immunoprecipitation: optimization, quantitative analysis and data normalization." Plant methods 3.1 (2007): 11.
*
Harb, Hani, et al. "Epigenetic regulation in early childhood: a miniaturized and validated method to assess histone acetylation." International archives of allergy and immunology 168.3 (2015): 173-181.

2)     Is there any explanation for the unchanged levels of Ova sIgE and mast cell degranulation? Has this phenomena been observed before in similar studies?

For OVA-specific IgE levels, we know from other studies that there can be some variation between animals*. Unfortunately, because of this variation the difference between the raw milk and shop milk group did not reach significance. Since the acute allergic skin response is the primary outcome parameter of this study, the power calculation for the number of animals in each group was based on this response. In addition, we showed before that changes in the allergic response do not always correlate with changes in allergen-specific IgE**. Both aspects may have contributed to the non-significant changes in IgE. We cannot explain why we did not observe any effect on mMCP-1 but timing of serum sampling after challenge might be of influence.
*
Hodgkinson, A., McDonald, N., & Hine, B. (2014). Effect of raw milk on allergic responses in a murine model of gastrointestinal allergy. British Journal of Nutrition, 112(3), 390-397. doi:10.1017/S0007114514001044
* van Esch, Betty CAM, et al. "Post‐sensitization administration of non‐digestible oligosaccharides and Bifidobacterium breve M‐16V reduces allergic symptoms in mice." Immunity, inflammation and disease 4.2 (2016): 155-165.

** Schouten, Bastiaan, et al. "Cow milk allergy symptoms are reduced in mice fed dietary synbiotics during oral sensitization with whey." The Journal of nutrition 139.7 (2009): 1398-1403.

3)     Out of curiosity, why was only acetylation of histones analysed as a marker of epigenetic changes and not the other modifications?

We are very grateful for this remark. First, we selected histone acetylation due to its substantial contribution to the regulation of the genomic DNA accessibility to the transcriptional machinery, i.e. chromatin opening. Second, histone acetylation is at the same time a good marker of the overall/general chromatin opening status*.

We have expanded the relevant sentence as follows: “Since histone acetylation substantially contributes to and is an important marker for an open chromatin structure [Alashkar Alhamwe et al., 2018; Potaczek et al., 2017], …” (page 10, line 410-412).

* Alhamwe, Bilal Alaskhar, et al. "Histone modifications and their role in epigenetics of atopy and allergic diseases." Allergy, Asthma & Clinical Immunology 14.1 (2018): 39.
*
Potaczek, Daniel P., et al. "Epigenetics and allergy: from basic mechanisms to clinical applications." Epigenomics 9.4 (2017): 539-571.

4)     From the outcomes of this study, especially with the epigenetic changes before and after challenge, would you be able to comment how this can translate into human milk consumption that may have protective effects against food allergy?

Thank you very much also for this stimulating comment. Once cannot be sure but indeed, it is quite possible that the protective effects of human breast milk consumption against food allergy could be, at least partially, attributed to the mechanisms similar to those observed in our study.

The reasons for anti-allergic effects of human breast milk seem to be complex. Human breast milk contains not only nutritional components but also functional molecules such as polysaccharides, cytokines, proteins and other components forming a real biological system which has the ability to modulate and shape the innate and adaptive immune responses of the infant in very early life*. How those components affect the epigenetic status of the growing child needs to be addressed in future studies.

We have included a respective comment: “Possible involvement of the epigenetic mechanisms should be investigated also in case of the anti-allergic effects of human breast milk consumption [Rajani et al., 2018].” (page 11, line 460-461).

* Rajani, Puja Sood, Antti E. Seppo, and Kirsi M. Järvinen-Seppo. "Immunologically active components in human milk and development of atopic disease, with emphasis on food allergy, in the pediatric population." Frontiers in Pediatrics 6 (2018): 218.
*
Ballard, Olivia, and Ardythe L. Morrow. "Human milk composition: nutrients and bioactive factors." Pediatric Clinics 60.1 (2013): 49-74.Round  2

Reviewer 2 Report

I have no further comments.